# Factors associated with depression among heart failure patients at selected public hospitals in Addis Ababa, Ethiopia: A cross sectional study

**Kassahun Alemayehu, Yohannes Ayalew Bekele, Teshome Habte Wurjine** *

School of Nursing and Midwifery, College of Health Sciences, Addis Ababa University, Addis Ababa, Ethiopia

* teshome.habte@aau.edu.et

**Data Availability Statement:** All relevant data are included within the manuscript document.

## Abstract

This study aimed to Assess Factors associated with depression among heart failure patients at cardiac follow-up clinics in a government teaching hospital of Addis Ababa. A cross-sectional study design was employed to assess Factors associated with depression among 424 heart failure patients at selected public hospitals of Addis Ababa who were selected by using a systematic random sampling method from January 1 to 30, 2021 at four public hospitals. Sample was proportionally allocated for each study hospital and then data were collected by using structured-interview questionnaires. Bivariate and Multivariate logistic regression analysis was done to examine the possible predictors and variables with the statistical association of P-value of < 0.05 and a 95% confidence interval were considered. Data were gathered from heart failure patients in cardiac follow clinic with 100% response rate. Among the 424 respondents [mean age: 52.7 (SD) 17.5 years; 56.1% women], prevalence of depression was 56.1%. Among the 424 respondents [mean age: 52.7 (SD) 17.5 years; 56.1% women], prevalence of depression was 56.1%. New York Heart Association class III and IV was highly associated with depression respectively. Furthermore, poor self-care behaviours alcohol use, poor social support, poor knowledge level, were associated with depression and statistically significant. The findings from this study showed that depression is highly prevalent among heart failure patients and age of respondent, alcohol intake, self-care behaviour, social support, knowledge level, and co-morbidity were independently associated with depression. Therefore, all institutions of cardiac centre should work on screening for depression in heart failure patients, and consult psychiatrists and psychologists for early detection and measures.

## 1. Introduction

According to Global Burden of disease report, 62 million people suffered from problem related to heart failure and more than half of the patients were in severe stages. Depression is one of

**Funding:** The authors received no specific funding for this work.

**Competing interests:** The authors have declared that no competing interests exist.

the most co-morbid disorders in patients with cardiovascular disease and it is one of the main public health problems across the world [1,2].

A study indicates that depression has strong association with heart failure (HF). Depressed patients have less functionality with increased physiological activity of the heart, heart failure symptoms and impaired health-related quality of life. Moreover, HF patients with depression are at risk for re-hospitalization [3]. Therefore, HF with depression, highly influence on their quality of life and they had 2-fold increased risk of death or cardiac events. mortality rates at five years for HF patients is 50% [4]. Among heart failure patients with depressive symptoms there will be an increased risk of mortality 2–3 times higher than in patients without symptoms of depression [5]. Furthermore, hospitalized heart failure patients with depression are at particularly at high risk for mortality. Median survival is 1.7 years for men and 3.2 years for women, with only 25% of men and 38% of women surviving for the last 5 years. This mortality rate is 4–8 times greater than that of the general population with similar age [6,7]. A study done in New York Hospitalized Heart Failure (HF) patients with depression the rate from 13% to 77.5% and out- patients are from 13% to 42% and depression is five times more prevalent in HF patients compared to the whole population [8,9]. Another study conducted in west Amhara region in Ethiopia, the incidence of depression among hear failure patients is 49%, and Dessie were 50% [1,10].

The presence of social support has been associated with lower incidence of depression and faster remission of depressive symptoms. In contrast, the lack of social support, conflict relation-ship have been linked to the presence of depression [11]. A systematic and meta-analysis conducted in Ethiopia indicates that depression is a common co-morbid illness among patients with diabetes [12]. And another meta-data analysis done in Greece, Patients classified in New York Heart Association (NYHA) in class "I to IV" of them class III and IV were more likely to be depressed than class I and II Patients [13].

According to the studies done in Nigeria and Greece, depression was associated with poor health care behaviors and additional risks factors, such as smoking, sedentary life, unhealthy dietary habits, lack of regular exercise, uncontrolled weight gain may lead to worsening of depression [14,15]. Another study done in Brazil indicates that selfcare behaviour was significantly associated with depression of the heart failure participants [16].

A cross sectional study conducted in west Amhara region in 2019 showed that poor knowledge of HF patients has strong association with depression [1,17]. But factors associated with depression in heart failure are not assessed adequately in developing countries including Ethiopia. Therefore, this study aimed to assess Factors associated with depression among heart failure patients at selected public hospitals in Addis Ababa, Ethiopia. This study finding may support for health care professionals to focuses their interventional strategies on the management of depression in heart failure patients, policy makers and responsible offices at various level of health care interventions to take appropriate measures and serve as a base line information for other researchers who are interested to conduct similar studies.

## Conceptual frame work

Conceptual framework for this study was established after reviewing and adapted from different literatures related to similar sociodemographic characteristics of the study population and identified variables as illustrated in the "*Fig 1*" below, the socio-demographic factors (marital status, age, gender of the participants, educational level and occupation), self- care behavior such as smoking habit and alcohol use, psycho-social factor like cognitive, perception and social support and co-morbidity like heart failure patients with diabetic, hypertension and chronic kidney disease and stage of heart failure and associated factors of depression. The

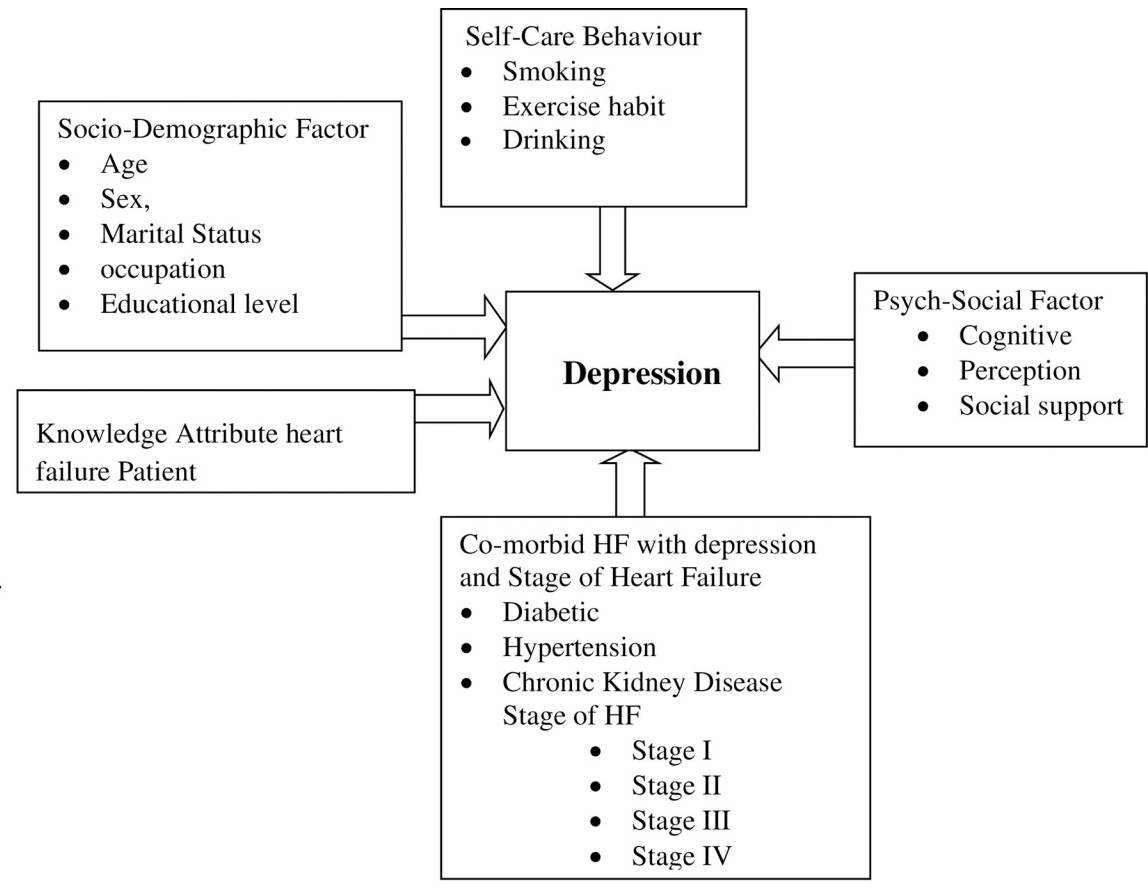

Figure 1, shows the socio-demographic factors (marital status, age, gender of the participants, educational level and occupation), self- care behaviour such as smoking habit and alcohol use, psycho-social factor like cognitive, perception and social support and co-morbidity like heart failure patient with diabetic, hypertension and chronic kidney disease and stage of heart failure and associated factors of depression. The direction of the relationship between outcome variable (Depression) and explanatory variables are illustrated.

**Fig 1. Conceptual framework for assessment of prevalence and associated factors of depression among heart failure patients at cardiac follow-up clinic.**

direction of the relationship between outcome variable (Depression) and explanatory variables are illustrated [3,12,18].

Government Hospitals with cardiac centre found in Addis Ababa are: $X_1$, $X_2$, $X_3$ and $X_4$ hospitals $X_1$ (1667)$X_3$ (200) $n_1$=246$n_4$=25$n_{total}$ =$n_1$+$n_{2+}$ $n_{3+}$ $n_4$ =246+123+30+25=**424**Lottery methodProportional allocation

### Ethical approval and consent to participate

Ethical clearance and approval were obtained from Institutional Review Board (IRB) of Addis Ababa University, College of Health Sciences and approved by Department of Nursing and Midwifery. Permission was obtained from clinical director of each study hospitals, Matron and heads of the respective ward. After explaining the purpose of the study, possible benefit of the study and time to complete the questionnaire and why the participants are chosen, oral and written informed consent was obtained from each participant before proceeding the procedure. The participants were fully explained that they have the right not to participate in the study, to stop at any time in between or not to answer any questions they were not willing to answer. Confidentiality was maintained; no unauthorized person had access to the information and names or other identifiers were not recorded. All methods and subjects provided written consent, and the study was conducted in accordance with Helsinki declarations.

## 2. Methods and materials

### 2.1. Study setting

The study was conducted at selected public hospitals of Addis Ababa city administration, Ethiopia. Addis Ababa is the largest and the most populated capital city of Ethiopia. It is a metropolitan area with a population of estimated to be around 5,006,000 people in 2021. The capital city holds 527 square kilometers of area in Ethiopia. The population density is estimated to be near 5,165 individuals per square kilometer available. Based on the 2020 population enumeration annual growth rate is 4.42%. The city has 15 public hospitals from this four of them randomly selected to conduct this research study namely: Tikur Anbesa specialized hospital, St. Paulo's hospital, Yekatit 12 Hospital and Armed force specialized hospital.

A. ***Tikur Anbesa Specialized Hospital (TASH):*** built by empire Hailesilasie. TASH in1972 the hospital become the only site for training medical doctor. in 1998 the TASH the largest referral hospital in the country with 850 beds. TASH now the main teaching center for both clinical and preclinical training of most discipline and it has 200 doctors, 379 nurses, and also have 950 permanent and contract administrative staffs to support hospital activity.

B. ***St. Paulo's Hospital***: established in 1969. It has 350 beds annually average of 300. 000 patients flow and.it has more than 5 million catchment population. The hospital has 1200 clinical and non-clinical staffs.

C. ***Yekatit 12 Hospital***: it is under the governance of Addis Ababa city administrative health bureau. The hospital provides service for population approximately 4 million people. It has 9 departments and 6 units with 265 beds, it is main referral hospital for treatment of burn. and the burn unit has 19 beds, 12 for adult and 7 for pediatrics.

D. ***Armed Force Specialized Hospital***: formerly known as prince Tsehay Memorial Hospital. It changes the name after 1974 revolution. It has 350 beds, 150 doctors 300 nurses 100 administrative staffs [19,20].

### 2.2. Study design and period

An institution-based cross-sectional study design was conducted from January 1 to 30, 2021.

### 2.3. 2.3. Study population

All heart failure patients those who had follow- up at selected four government hospitals of cardiac centre in Addis Ababa during the study period.

## 2.4. Sampling procedure and technique

Those patients who were clinically confirmed as having heart failure assessed by reviewing their clinical charts with a check list that incorporates the PHQ-9 tool for depression measurement used. And patients assessed using structured-interview questionnaires and to select a total of 424 heart failure patients sampled from 2,867 heart failure patients after proportionally allocated patients from four selected public hospitals as stated below.

The total number of heart failure patients in the four selected study hospitals were 2,867. Hence, after the sample size was determined by using simple population proportion formula (n-424). The study sample was proportionally allocated to each study hospitals in line with: proportional allocation formula.

- **Tikur Anbesa specialized hospital ($X_1$)**: the total number of heart failure patients in this hospital was = 1667, proportional allocation of sample was calculated n = 1667×424/2867 = **246.**

- **St Paulo's specialized hospital ($X_2$)**: the total number of heart failure patients in this hospital was = 833 proportional allocation of sample was calculated n =, 833×424/2867 = **123.**

- **Armed force specialized hospital ($X_3$):** the total number of heart failure patients in this hospital was = 200 proportional allocation of sample was calculated n = 200×424/2867 = **30.**

- **Yekatit 12 hospital ($X_4$)**: the total number of heart failure patients in this hospital was = 167 proportional allocation of sample was calculated n = 167×424/2867 = **25.**

The sample size was proportionally allocated for each hospital as depicted in "*Fig 2*" below. Study participants were selected from proportionally allocated study subject in each hospital using a random sampling technique from eligible patients visiting the cardiac clinic during the data collection period was interviewed.

## 2.5. Procedure of Data Collection and tool used

Data was collected by four trained BSc nurses using 5% pretested interviewer administered questionnaire and supervised by two MSc nurses. The data collection instrument includes the following components.

**PHQ**-9: by using a check-list that was developed on the basis of prior similar studies, the data was collected by using the Patients Health Questionnaire (PHQ-9), the questionnaire has nine items, the total score ranges from 0 to27 a score 5,10,15,20 represent cut point for mild, moderate, moderate-severe and patient e depression respectively. In PHQ-9 tool there are four options (0 = none at all, 1 = several days, 2 = more than half of the days, and3 = nearly every day). Which have been used to screen depression symptoms from the study participants 21. and it has 88% specificity and sensitivity. HF patients score between 1–4 categorized as having "no depression". HF patients score between 5–9 was "mild depression", HF patients score between 10–14 was "moderate depression", between 15–19 categorized as having "moderate to severe depression" and score >20 was categorized as having "severe depression". PHQ-9 tool Cronbach alpha value was .904 [**21,22**].

**European HF Self-care behaviours scale-9 (EHFScBC-9)**: The EHFScBS-9 had supportive psychometric properties of validity, reliability and precision, and it's used to measure self-care behaviours in clinical practice and research. The EHFScBS-9 has nine items of questions each item uses a 5-point Likert scale from 1 ("completely agree") to 5 ("completely disagree"). The possible score was 9 to 45, with the level of self-care behaviour score described as the following: A score <or = 2 on each item and total score < or = 18 suggest high self-care behaviours, a

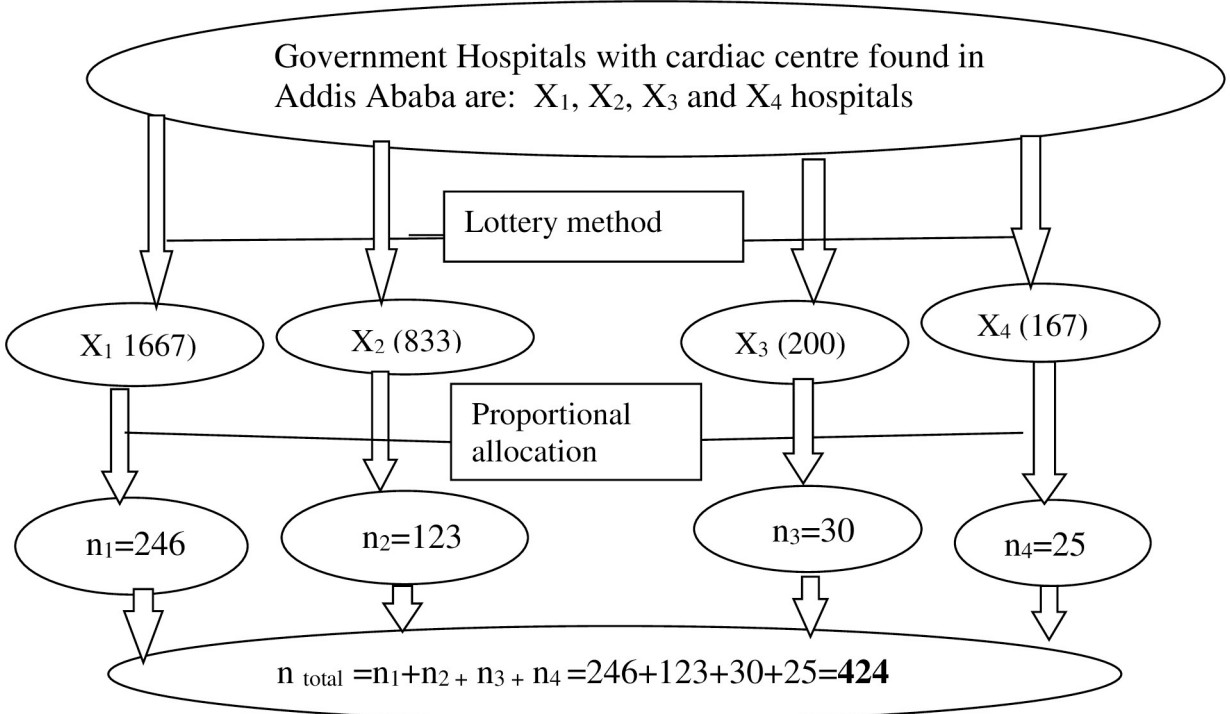

*Figure 2 above illustrates* the sample size was proportionally allocated for each hospital. Study participants were selected from proportionally allocated study subject in each hospital using a random sampling technique from eligible patients visiting the cardiac clinic during the data collection period was interviewed.

**Fig 2. Schematic presentation of sampling technique and sample size proportional allocation.**

score >2 on each item and total score >18 suggest low self-care behaviours) [22,23]. Alpha Cronbach was 0.806.

**Oslo social support scale (OSSS-3):** Evidence supports reliability and validity of the OSSS-3 as a measure of social determinants of health in the general population. The OSSS-3 consists of three items assessing the level of social support the sum score ranges from 3 to 14, with high values (>8) representing strong levels and low values (3–8) representing poor levels of social support [14]. Alpha Cronbach in this study was 0.869.

**The Dutch HF knowledge scale** has 15 multiple choices for each item patients can choose from the three options, with one of the options being the correct answer. When a person gives the correct answer, 1 point is given whereas the answer is wrong the person receive 0 point for that question. The possible total score for knowledge of heart failure ranges from 0 to 15 and interpreted as Study subject's knowledge on HF was found to be a score of DHFKS above Median was used as a cut off for good knowledge and poor knowledge as the score is below Median [24].

## 2.6. Data quality assurance

Structured and pre-tested questionnaire was used and also training was given for data collectors and supervisors on the objective of the study, method, contents and also how to maintain confidentiality and privacy of the study subject was strictly conducted. Data was collected by four experienced staff nurses with BSc degree and above. Pre-test was conducted on 5% of heart failure patients at Zewditu hospital before the actual data collection period and based on the finding the necessary correction made on the method and materials.

## 2.7. Data processing and analysis

The collected data was entered in to epi data processed and analysed by using SPSS version 25. Descriptive statistics was employed to describe the percentages and frequency distributions of the variables in the study. Adjusted odd ratio with 95% confidence interval was estimated to measure the strength of association a P- Value of $\leq 0.05$ was used for statistical significance.

## 2.8 Operational definitions

*Depressive symptom*-The Patients Health Questionnaire 9 (PHQ9) is a self-questionnaire designed to help diagnosis and detect mental disorders commonly encountered in the primarily clinical setting. The individual participant had interviewed depressive symptoms when he/she has at least four positive symptoms in the PHQ-9 interview. including Q1(Little interest or pleasure in doing things) and Q2 (Feeling down, depressed or hopeless) [25,26].

  **Major depressive disorder**: A score of $\geq 10$ on the PHQ-9 scale was considered as a major depressive disorder [27].

  *Poor self-care*-The individual participant had poor self-care when scored with from 9 to 45 with a higher score above >18 indicating poor, and lower <18 is good self-care of European Heart Failure Self-Care Behaviour Score-9 [17].

  *Poor social support*-The individual participant had poor social support when she/he scored between the 3–8 out of 14 [23].

  *Knowledge*: A score of DHFKS above Median was used as a cut off for good knowledge and poor knowledge as the score is < Median [28].

  *Knowledge of HF* patients defines as ability to recognize and interpreted HF symptoms [29].

  *Low knowledge*- A score of DHFKS "$\geq 10$" was used as a cut off for good knowledge and poor knowledge as the score is "<10" [28].

  *Self–care behaviour*: is defined as the activity of HF patients performs to take care of their health in terms of exercise habit, abstain from cigarette smoking and alcohol intake [17].

  *Psycho-social support*: defines as help maintain a continuum of social support during or after a problem and prevent from long term mental disorder and make him to developed good perception [23].

  *Co-morbidity*-The individual heart failure patients has additional chronic disease like DM, HTN, CKD [17].

  *Diabetes mellitus*: -a disease in which the body's ability to produce or respond to the hormone insulin is impaired, resulting in abnormal metabolism of carbohydrate and elevated level of glucose in the blood [30].

  *Hypertension*: -abnormal high blood pressure which is systolic blood pressure >130 and diastolic blood pressure >85mmHg [25].

  *Chronic kidney disease*: is abnormality of kidney structure or function, present for >3 month, with implications for health [30].

## 3. Result

### 3.1 Socio-demographic characteristics

A total of 424 study participants were included with 100% response rate and about 238 (55.2%) of the study participants were female. The mean and standard deviation of age of the respondents were 52.68 and17.471years respectively. A high proportion of 134 (31.6%) of the respondents were within the age group of 37–54 years and about 166 (39.1%) the study reveals participants were married. The study found that one out of four participants did not attend formal education or illiterate and two-third of participants were employed in government and non-governmental institutions. As shown in *Table 1* below. About 36.3% participants are having a monthly income in the range of 601–1650 Ethiopian birr.

### 3.2 Clinical characteristics of respondents

From the total of 424 Heart failure patients about 415 (97.9%) of them found to have comorbidity. According to the New York Heart Association guideline more than one third of the participants 140(33%) were categorized in class III, patients experienced hypertensive disorder 147(34.7%), diabetes mellites 109(25.7%), and chronic kidney disease accounts 19 (4.5%).

**Table 1. Socio-demographic characteristics of heart failure patients attending cardiac follow-up clinic *at selected government hospitals* in Addis Ababa, Ethiopia.** (n = 424).

| Socio-demographic factor | Category | Frequency | Percentage |
|---|---|---|---|
| Gender | Male | 186 | 43.9% |
| | Female | 238 | 56.1% |
| Age | 18–36 | 100 | 23.6% |
| | 37–54 | 134 | 31.6% |
| | 55–72 | 114 | 26.9% |
| | >72 | 76 | 17.9% |
| Marital status | Married | 166 | 39.1% |
| | Divorced | 117 | 27.6% |
| | Single | 61 | 14.4% |
| | Widow | 80 | 18.9% |
| Educational level | Illiterate | 110 | 25.9% |
| | Elementary | 104 | 24.5% |
| | Secondary | 115 | 27.1% |
| | Preparatory | 40 | 9.4% |
| | University level | 55 | 23.8% |
| Profession | Governmental employ | 101 | 23.8% |
| | Self-employ | 175 | 41.3% |
| | Pension | 18 | 4.2% |
| | Student | 54 | 12.7% |
| | House wife | 76 | 17.9 |
| Monthly income per month | 0–600 | 9 | 2.1% |
| | 601–1650 | 154 | 36.3% |
| | 1651–3200 | 67 | 15.8% |
| | 3201–5250 | 69 | 16.3% |
| | 5251–7300 | 84 | 19.8% |
| | 7301–10899 | 29 | 6.8% |
| | >10899 | 12 | 2.8% |

### 3.3 Self-care behaviour, social support and knowledge of respondents

Majority of participants 257(60.6%) were having poor knowledge and more than half 230 (54.2%) of study participants had poor social support. The mean of heart failure patients with self-care behaviour score was 19.29, with SD of 4.91. From the total study participants,235 (55.4%) had poor self-care behavior with cigarette smoking 120 (28.3%). More than half 235 (54.4%) of them were alcohol users.

### 3.4 prevalence of depression

The study indicates that depression is common problem as ***shown in Fig 3*** below, the incidence of depression was 59% (95% CI 54.5–63.7). Being in NYHA class III and IV [(AOR:12.8 (2.2–71.6),95%CI, P = **0.004, (**AOR:19.2(1.9–189.9), 95% CI, P = **0.011)**] respectively, having poor self-care behaviours[(AOR: 9.1(2.4–34.6), 95%CI, P = **0.001)**], having alcohol use[(AOR: 17.7(4.14–35.65),95% CI,P = **0.001)**], having poor social support [(AOR: 4.6(1.2–16.7),95%CI, P = **0.020**)], having poor knowledge [(AOR:5.1(1.3–20.6), 95%CI, P = **0.020**), and being single or unmarried [(0.108(0.03–0.47), 95%CI, P = **0.001**)] were independently associated with depression. Low educational attainment or being illiterate was also associated with depression.

### 3.5 Factor associated with depression among heart failure patients attending cardiac follow-up clinic

The bi-variate logistic regression analysis indicates that, age of respondent, marital status, educational status, profession, NYHA classification, self-care behaviour, high alcohol intake, social

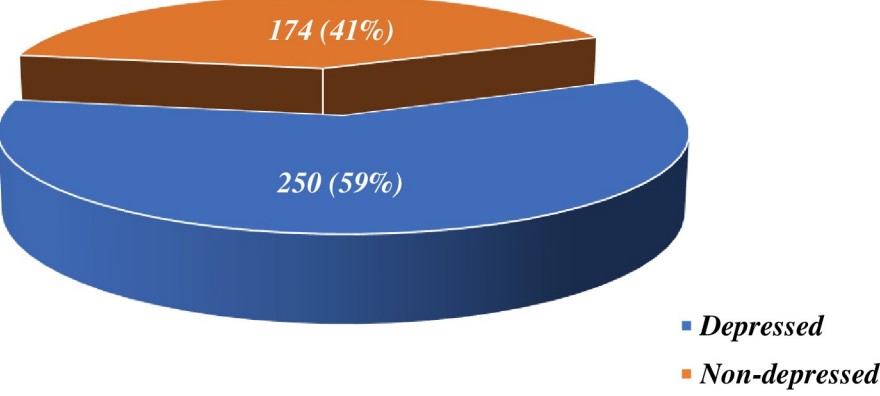

Figure 3 above shows the prevalence of depression was 59% and it is statistically significant with (95%CI 54.5-63.7), being New York Heart Association (NYHA) class III and IV [(AOR:12.8 (2.2—71.6), 95%CI, P=0.004, (AOR:19.2(1.9-189.9),95%CI, P=0.011)] respectively.

**Fig 3. Prevalence of depression among heart failure patients attending cardiac follow-up clinic at selected government specialized teaching hospitals in Addis Ababa, Ethiopia (n = 424).**

**Table 2. Factor associated with depression in bi-variate logistic regression among heart failure patients attending cardiac follow-up clinic *at selected government hospitals* (n = 424).**

| Variable | Category | Depressive symptom | | COR | P -Value |
|---|---|---|---|---|---|
| | | Non depression | depression | | |
| Marital status | Single or un-married | 53(30.5%) | 205(82%) | 10.4(6.5–16.4) | <0.001 |
| | Married | 121(69.5%) | 45(18%) | 1 | |
| Age | 18–36 | 76(43.7%) | 24(9.6%) | 1 | |
| | 37–54 | 86(49.4%) | 48(19.2%) | 1.7(0.99–3.1) | 0.054 |
| | 55–72 | 8(4.6%) | 106(42.4%) | 41.9((17–98.4) | 0.001 |
| | >72 | 4(2.3%) | 72(28.8%) | 57(18.8–172.3) | <0.001 |
| Educational level | Illiterate | 13(7.5%) | 97(38.8%) | 4.8(1.-12.) | <0.001 |
| | Higher school | 125(71.8%) | 134(53.6%) | 0.874(0.41–1.86) | 0.728 |
| | University level | 36(20.6%) | 19(7.6%) | 1 | |
| Profession | Governmental employ | 77(44.3%) | 24(9.6%) | 7.4(4.4–12.5) | <0.001 |
| | Non-governmental employ | 97(55.7%) | 226(90.4%) | 1 | |
| NYHA Class | I | 75(43.1%) | 17(6.8%) | 1 | |
| | II | 69(39.7%) | 46(18.4%) | .019(0.007–0.051) | <0.001 |
| | III | 24(13.8%) | 116(46.4%) | .056(.023-.140) | <0.001 |
| | IV | 6(3.4%) | 71(28.4%) | .408(.159–1.048) | 0.068 |
| Co-morbidity | Diabetes mellites | 26(14.9%) | 83(33.2%) | 5.9(3.4–10.3) | <0.001 |
| | Hypertension | 38(21.8%) | 109(43.6%) | 5.3(3.2–8.8) | <0.001 |
| | CKD | 13(7.5%) | 6(2.4%) | 0.81((0.3–2.3) | 0.775 |
| | Non-co-morbid | 97(55.7%) | 52(20.8%) | 1 | |
| Alcohol intake | Yes | 24(13.8%) | 211(84.4%) | 33.8(19.5–58.6) | <0.001 |
| | No | 150(86.2%) | 39(15.6%) | 1 | |
| Self-care behaviour | Good | 163(93.7%) | 26(10.4%) | 1 | |
| | Poor | 11(6.3) | 224(89.6%) | 127.6(61.3–265.7) | <0.001 |
| Social-support | Good | 158(90.8%) | 36(14.4%) | 1 | |
| | Poor | 16(9.2%) | 214(85.6%) | 58(31.4–109.5) | <0.001 |
| Knowledge | Good | 143(82.5%) | 24(9.6%) | 1 | |
| | Poor | 31(17.8%) | 226(90.4%) | 43.3(24.5–77) | <0.001 |

N.B COR = crude odd ratio, AOR = adjusted odd ratio, Multivariate statistical analysis was applied.

support, knowledge level of the disease prognosis and co-morbidity were associated with depression among heart failure patients as shown in *Table 2* below.

The multivariate logistic regression analysis also shows that, NYHA class, self-care behavior, high alcohol intake, social support, knowledge level and marital status were found to be associated with depression and statistical significance with a P-value of less than 0.05.

Those heart failure patients who had NYHA Class category of III and IV were more likely to be depressed than class I and II[(AOR:12.8(2.2–71.6), 95% CI, P = **0.004, (**AOR:19.2(1.9–189.9), 95% CI, P = **0.011)**] respectively. Those heart failure patients who had poor self-care behaviour are 9-fold at risk of depressive disorder as compared to those who had good self-care behaviour [(AOR: 9.1(2.4–34.6),95% CI, P = 0.001)]. Those heart failure patients they were alcohol users are 18 times more likely to be depressed than those who have not taken alcohol [(AOR: 17.7(4.14–35.65),95% CI, P = **0.001)**]. Those heart failure patients who were poor social support are 5 times more likely to be depressed when compared to good social support [(AOR: 4.6(1.2–16.7),95%CI, P = **0.020**)] as depicted in *Table 3* below. In this study those

**Table 3. Factor associated with depression in multi-variate logistic regression among heart failure patients (n = 424).**

| Variable | Category | Depressive symptom | | AOR | P Value |
|---|---|---|---|---|---|
| | | Non Depression | Depression | | |
| Marital status | Single or not Married | 53(30.5%) | 205(82%) | 0.108(0.03–0.47) | **0.001**\* |
| | Married | 121(69.5%) | 45(18%) | 1 | |
| NYHA Class | I | 75(43.1%) | 17(6.8%) | 1 | |
| | II | 69(39.7%) | 46(18.4%) | 3.8(0.81–17.9) | 0.09 |
| | III | 24(13.8%) | 116(46.4%) | 12.8(2.2–31.6) | **0.004**\* |
| | IV | 6(3.4%) | 71(28.4%) | 19.2(9–89.9) | **0.011** |
| Alcohol intake | Yes | 24(13.8%) | 211(84.4%) | 17.7(4.14–35.65) | **<0.001**\* |
| | No | 150(86.2%) | 39(15.6%) | 1 | |
| Self-care behaviour | Good | 163(93.7%) | 26(10.4%) | 1 | |
| | Poor | 11(6.3) | 224(89.6%) | 9.1(2.4–34.6) | **0.001**\* |
| Social-support | Good | 158(90.8%) | 36(14.4%) | 1 | |
| | Poor | 16(9.2%) | 214(85.6%) | 4.6(1.2–16.7) | **0.020**\* |
| Knowledge | Good | 143(82.5%) | 24(9.6%) | 1 | |
| | Poor | 31(17.8%) | 226(90.4%) | 5.1(1.3–20.6) | **0.020**\* |

N.B COR = crude odd ratio, AOR = adjusted odd ratio.

heart failure patients who had poor knowledge were more likely 5 times to be depressed as compared to good knowledge[(AOR:5.1(1.3–20.6),95%CI, P = 0.020)]. Those heart failure patients who are unmarried 9.25 times more likely to be depressed than married [(0.108(0.03–0.47),95%CI, P = **0.001**)].

## 4. Discussions

The aim of this study was to assess Factors associated with depression among heart failure patients at selected public hospitals in Addis Ababa, Ethiopia. The incidence of depression in this study reveals that 59% (95% CI,54.5–63.7), this indicates that depression is highly associated with heart failure. It is obviously clear that depression is highly affecting the quality of life and efficacy of care.

A study conducted in Australia shows the incidence of depression was 52% and South Africa in Johannesburg indicates 50% [8,21]. This similarity might be due to problem of depression in heart failure patients spreading across in different nations globally. The current study reveals that higher incidence compared to study done in Greece (20–40%), Japanese (5.8%), United States of America (42.1%) [20,27,28]. The difference might be due to different screening strategy, study design and sociodemographic characteristics of the study population. And incidence of depression observed in the present study was higher than a study conducted in west Amhara region of Dessie city administration were 50% [1], this variation might be different educational background, their life style and sociodemographic characters.

This study indicated that those heart failure patients with NYHA class III and IV are more likely at risk of depressive disorders compare to those who had NYHA class I and II. The possible explanation could be individual heart failure patients with advanced stage might be worry about their worsening symptom, illness-related complication, dietary restriction and un able to do any activity and they are always dependent on others. This might be directly or indirectly lead to depression. This finding supported by a study done in Greece 2020, and study done in Ethiopia [1,15]. This indicate that advanced heart failure patients need to be early evaluation of depression and cardiac clinic work in collaboration with psychiatry department to screen

and therapeutic interventions. This study found that heart failure patients who had poor self-care behaviour is positively associated with depression when compared to good self-care behaviour, this might be due to poor self-care behaviour, prone to depression and potential to develop bad habit, like cigarette smoking, the use of shisha, chat chewing and lack of regular exercise. The other co-relation between psychological factors and disease outcomes, such as poor quality of life, effect of poor practice of self-care behaviour as the result heart failure patients may be potential to develop depression. This finding was in line with the study done in New-York, Brazil and Gondar [10,16,17]. This similarity might be due to prevention strategy of the country and pathological nature of disease process.

In this study, those heart failure patients who experienced high alcohol intake shows strongly positive association with depressive disorder than those who have not experienced alcohol use [(AOR: 17.7(4.14–35.65),95% CI, P = **0.001**)] this might be due to the fact that alcohol use exposes to depression. Most alcohol users might be affected by physiological, psychological, social and economic behaviour that can alters metabolic condition of an individual life and as the result of conflict to family and society at large. This finding supported by research done in England, [24] and this finding contradict to research done in United Kingdom showed that there is no significant association alcohol intake and depression. The difference might be due to difference in socio-demographic characteristics of the study population. And also, this study indicates that heart failure patients who had poor social support is more likely to be depressed when compare to good social support [(AOR: 4.6(1.2–16.7),95%CI, P = **0.020**)]. The reason might be patients who have poor social support may not share their own stressor. and also plays great role in the coping strategies, so that this situation might be directly or indirectly expose to depression. This finding supported by research done in Pakistan, USA and Greece [25,26]. This reveals that and contact with support group for those HF patients who had poor social support. In this study heart failure patients who had poor knowledge about their disease were having positive associated with depression [(AOR:5.1(1.3–20.6),95%CI, P = **0.020**)]. This finding supported by research done in Ethiopia [1,30]. This indicate that health care institution should be focused on health education specially for heart failure patients who had poor knowledge. In general health care professionals should focus on education about their disease process and associated factors.

This study reveal that unmarried patients have positively associated with depression compared to married ones. [(AOR:0.108(0.03–0.47),95%CI, P = **0.001**)] This might be heart failure patients who were unmarried did not share their own stressor to life partner. This finding was in line with the study conducted in Pakistan and Ethiopia [11,27].

## 5. Conclusion and recommendations

### 5.1 Conclusion

This study reveals that Factors associated with depression among heart failure patients at selected public hospitals found to be very high and advanced stage of heart failure patients were more depressive, poor self-care behaviour, alcohol users and poor social support were more likely at risk of depressive disorder. And also, heart failure patients who had good knowledge are at lower risk of depressive disorder and having single or unmarried patients were at risk of depressive disorder and those variables were associated with the odds of depression among heart failure patients in Addis Ababa.

### 5.2 Recommendations

All health institution of cardiac units should work on screening of heart failure patients for depression and consult psychiatrist and Psychologist for early detection and possible measure.

In addition to this health care workers should focus to teach heart failure patients about disease prognosis and associated risk factors and patients' education should be a part of heart failure patients management guideline.

## Acknowledgments

The authors would like to acknowledge to Addis Ababa University, College of Health Sciences, School of nursing and midwifery and study participants for their unreserved support during data collection.

## Author Contributions

**Conceptualization:** Yohannes Ayalew Bekele, Teshome Habte Wurjine.

**Data curation:** Yohannes Ayalew Bekele, Teshome Habte Wurjine.

**Formal analysis:** Kassahun Alemayehu.

**Investigation:** Kassahun Alemayehu.

**Methodology:** Kassahun Alemayehu.

**Project administration:** Kassahun Alemayehu, Teshome Habte Wurjine.

**Software:** Yohannes Ayalew Bekele.

**Supervision:** Yohannes Ayalew Bekele, Teshome Habte Wurjine.

**Validation:** Yohannes Ayalew Bekele, Teshome Habte Wurjine.

**Visualization:** Teshome Habte Wurjine.

**Writing – original draft:** Kassahun Alemayehu.

**Writing – review & editing:** Teshome Habte Wurjine.

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
