## [Decision Letter · Decision Letter 0]

7 Feb 2022

PGPH-D-22-00002

Prevalence of depression and associated factors among heart failure patients at cardiac follow-up clinics in a government teaching hospital at Addis Ababa, Ethiopia; A cross sectional study

Thank you for submitting your manuscript to PLOS Global Public Health. After careful consideration, we feel that it has merit but does not fully meet PLOS Global Public Health’s publication criteria as it currently stands. Therefore, we invite you to submit a revised version of the manuscript that addresses the points raised during the review process.

We look forward to receiving your revised manuscript.

Kind regards,

Andreas K Demetriades, MBBChir, MPhil, FRCSEd, FEBNS.

Academic Editor

Journal Requirements:

1. Please ensure that the Title in your manuscript file and the Title provided in your online submission form are the same.

2. Please provide separate figure files in .tif or .eps format only.  Please ensure that all files are under our size limit of 20MB.  

For more information about how to convert your figure files please see our guidelines: Once you've converted your files to .tif or .eps, please also make sure that your figures meet our format requirements

3. Please update the completed 'Competing Interests' statement, including any COIs declared by your co-authors. If you have no competing interests to declare, please state "The authors have declared that no competing interests exist". Otherwise please declare all competing interests beginning with the statement "I have read the journal's policy and the authors of this manuscript have the following competing interests:"

4. Please amend your detailed Financial Disclosure statement. This is published with the article, therefore should be completed in full sentences and contain the exact wording you wish to be published.

i) Please include all sources of funding (financial or material support) for your study. List the grants (with grant number) or organizations (with url) that supported your study, including funding received from your institution. 

ii). State the initials, alongside each funding source, of each author to receive each grant.

iii). State what role the funders took in the study. If the funders had no role in your study, please state: “The funders had no role in study design, data collection and analysis, decision to publish, or preparation of the manuscript.”

5. Please ensure that the funders and grant numbers match between the Financial Disclosure field and the Funding Information tab in your submission form. Note that the funders must be provided in the same order in both places as well.

Additional Editor Comments (if provided):

Thanks for your submission to this journal.

The peer reviewers have provided some constructive critique which can hopefully help you in preparing a revised submission.

Reviewers' comments:

Reviewer's Responses to Questions

**Comments to the Author**

1. Does this manuscript meet PLOS Global Public Health’s publication criteria? Is the manuscript technically sound, and do the data support the conclusions? The manuscript must describe methodologically and ethically rigorous research with conclusions that are appropriately drawn based on the data presented.

Reviewer #1: Partly

Reviewer #2: Partly

2. Has the statistical analysis been performed appropriately and rigorously?

Reviewer #1: No

Reviewer #2: Yes

3. Have the authors made all data underlying the findings in their manuscript fully available (please refer to the Data Availability Statement at the start of the manuscript PDF file)?

Reviewer #1: Yes

Reviewer #2: Yes

4. Is the manuscript presented in an intelligible fashion and written in standard English?

Reviewer #1: No

Reviewer #2: No

5. Review Comments to the Author

Reviewer #1: The study investigated the prevalence of depression and associated factors among a random sample (424) of patients with heart failure enrolled from four public hospitals at Addis Ababa, Ethiopia.

Major comments:

- Data on heart failure from Ethiopia is scarce. Although revision of the manuscript had been undertaken for this round, the manuscript needs to be further edited for content, grammar and spelling. The authors should also pay attention to ensure correct and up-to-date references are cited.

- The authors should clarify how the diagnosis of heart failure was ascertained in the sample? Was a standard set of diagnostic criteria (e.g. Framingham criteria) used, combined with echocardiography? This should be clearly stated in the methods section.

- Ascertainment of depression, the use of Patient Health Questionnaire (PHQ-9, with the appropriate reference) to ascertain depressive symptoms should be clearly stated in first line of PHQ-9 on page 7. Reference for the PHQ-9 (21) is incorrect and should be replaced.

- Under sections on limitation, to include PHQ-9 Patient Depression Questionaire was self-administered and was not verified by the clinicians.

- Variables used for statistical adjustment (as in multivariable regression models) should be stated at bottom of tables. Similarly, variables/categories of variables used for statistical adjustment should be stated in the methods section – Data processing and analysis on Page 9.

- How are missing data handled?

- In the Discussion section, in addition to what has been written, the authors should discuss more about who the patients with depression/depressive symptoms are and where the disparities are, with the view to inform targets for surveillance or preventive measures. Clearly, from the findings, more women, illiterate or less educated, those lacking social support, etc, were more likely to manifest depressive symptoms. The findings have implications for social policy too.

Minor Comments:

- Title should be amended since patients were enrolled from four (not one) public hospitals;

- ‘New York heart association’ should be written as New York Heart Association on page 12, and more…(e.g.United States of America, not ‘United states of America’; United Kingdom, not ‘united kingdom’, etc).

Reviewer #2: Alemayehu et al, report on the prevalence of depression and associated factors among

heart failure patients at cardiac follow-up clinics in Ethiopia, addis ababa. This was a cross-section prospective study conducted in 4 hospitals in addis Ababa. The authors report that depression is prevalent among

heart failure patients and age of respondent, alcohol intake, self-care behaviour, social support,

knowledge level, and co-morbidity were associated with depression and found to be statistically significant. They then conclude that all cardiac care centres should work on screening for depression in heart

failure patients, with early consultation of psychiatrists and psychologists for early detection and measures.

This manuscript has a number of concerns that need to be reviewed in order to improve the overall quality of the paper.

1) The study population was preselected from an existing data base and these patients then interviewed within a 1 month period of January 2021. Its not clear how this data base was compiled / created. It is also not clear how the diagnosis of heart failure had previously been made? were all patients subjected to echocardiography ? and If so what were the mean ejection fraction in the cohort?

2) The randomly selected study participants were invited to participate in the study. Its not clear how long was their duration of living with heart failure ? Or when was the heart failure diagnosed and when did they participate in the study? This is important as acute and chronic heart failure patients will report differently to screening questionnaires

3) The aetiology of the heart failure diagnosis is also important. The authors only present the co-morbidity. A detailed heart failure aetiology needs to be reported on, as the underlying cause of heart failure has an important impact on the presence of depression symptoms

4) There is no report on the heart failure therapy that these patients were on. To better characterise the population. The respective therapy needs to be reported on.

5) The manuscript needs to be reviewed by a professional english ( scientific writing) editor. There were many grammatical errors. This would improve the overall quality of the manuscript.

6) Some of the contents in the manuscript need to be revised or reviewed. The introductory paragraph was very long, almost reading as though it was a literature review. This needs to be made more concise. The sections that speak to the various history behind the hospitals could be added as an appendix. The manuscript needs to be edited to fit the style of the journal original article manuscript and not a research report thesis.

7) A DSM IV criteria for diagnosing depression was not used. Therefore the authors need to talk about the presence of depressive symptoms and not a diagnosis of depression. This change needs to be made through-out the manuscript and consistently.

8) A study flow diagram would be beneficial as to how the total number of 424 was achieved form the database with 2867 participants.

9) In the introduction, I would counter argue that hypertension and dyslipidaemia are more prevalent than depression in global cardiovascular disease burden. Please review this statement.

6. PLOS authors have the option to publish the peer review history of their article (what does this mean?). If published, this will include your full peer review and any attached files.

**Do you want your identity to be public for this peer review?** For information about this choice, including consent withdrawal, please see our Privacy Policy.

Reviewer #1: No

Reviewer #2: **Yes: **Nqoba I Tsabedze

---

## [Decision Letter · Decision Letter 1]

5 May 2022

PGPH-D-22-00002R1

Prevalence of depression and associated factors among heart failure patients at cardiac follow-up clinics in a government teaching hospital at Addis Ababa, Ethiopia; A cross sectional study

Dear Dr. Wurjine,

Thank you for submitting your manuscript to PLOS Global Public Health. After careful consideration, we feel that it has merit but does not fully meet PLOS Global Public Health’s publication criteria as it currently stands. Therefore, we invite you to submit a revised version of the manuscript that addresses the points raised during the review process.

We look forward to receiving your revised manuscript.

Kind regards,

Andreas K Demetriades, MBBChir, MPhil, FRCSEd, FEBNS.

Academic Editor

Journal Requirements:

Additional Editor Comments (if provided):

Thank you for your revised submission

as you can see from the peer review there are still areas of potential improvement

Reviewers' comments:

Reviewer's Responses to Questions

**Comments to the Author**

1. If the authors have adequately addressed your comments raised in a previous round of review and you feel that this manuscript is now acceptable for publication, you may indicate that here to bypass the “Comments to the Author” section, enter your conflict of interest statement in the “Confidential to Editor” section, and submit your "Accept" recommendation.

Reviewer #1: (No Response)

Reviewer #2: (No Response)

2. Does this manuscript meet PLOS Global Public Health’s publication criteria? Is the manuscript technically sound, and do the data support the conclusions? The manuscript must describe methodologically and ethically rigorous research with conclusions that are appropriately drawn based on the data presented.

Reviewer #1: Partly

Reviewer #2: Partly

3. Has the statistical analysis been performed appropriately and rigorously?

Reviewer #1: Yes

Reviewer #2: I don't know

4. Have the authors made all data underlying the findings in their manuscript fully available (please refer to the Data Availability Statement at the start of the manuscript PDF file)?

Reviewer #1: Yes

Reviewer #2: Yes

5. Is the manuscript presented in an intelligible fashion and written in standard English?

Reviewer #1: No

Reviewer #2: Yes

6. Review Comments to the Author

Reviewer #1: Comments:

- The manuscript can be further improved with more editorial help for content. This version is however better than the previous, although it shouldn’t be accepted in this current form.

- Annotated suggestions/corrections have been added to the pdf, to assist the authors, since data from Ethiopia is scarce.

- Variables used for statistical adjustment (as in multivariable regression models) should be stated at bottom of tables.

- Title should be amended since patients were enrolled from four (not one) public hospitals (previous comment about this was made in last review).

Reviewer #2: This manuscript has a number of concerns that need to be reviewed in order to improve the overall quality of the paper.

1) The study population was preselected from an existing data base and these patients then interviewed within a 1 month period of January 2021. Its not clear how this data base was compiled / created. It is also not clear how the diagnosis of heart failure had previously been made? were all patients subjected to echocardiography ? and If so what were the mean ejection fraction in the cohort?

2) The randomly selected study participants were invited to participate in the study. Its not clear how long was their duration of living with heart failure ? Or when was the heart failure diagnosed and when did they participate in the study? This is important as acute and chronic heart failure patients will report differently to screening questionnaires

3) The aetiology of the heart failure diagnosis is also important. The authors only present the co-morbidity. A detailed heart failure aetiology needs to be reported on, as the underlying cause of heart failure has an important impact on the presence of depression symptoms

4) There is no report on the heart failure therapy that these patients were on. To better characterise the population. The respective therapy needs to be reported on.

5) The manuscript needs to be reviewed by a professional english ( scientific writing) editor. There were many grammatical errors. This would improve the overall quality of the manuscript.

6) A DSM IV criteria for diagnosing depression was not used. Therefore the authors need to talk about the presence of depressive symptoms and not a diagnosis of depression. This change needs to be made through-out the manuscript and consistently.

7) In the introduction, I would counter argue that hypertension and dyslipidaemia are more prevalent than depression in global cardiovascular disease burden. Please review this statement.

7. PLOS authors have the option to publish the peer review history of their article (what does this mean?). If published, this will include your full peer review and any attached files.

**Do you want your identity to be public for this peer review?** For information about this choice, including consent withdrawal, please see our Privacy Policy.

Reviewer #1: No

Reviewer #2: No

---

## [Decision Letter · Decision Letter 2]

13 Jul 2022

Factors associated with depression among heart failure patients at selected public hospitals in Addis Ababa, Ethiopia; A cross sectional study.

PGPH-D-22-00002R2

Dear 

We are pleased to inform you that your manuscript 'Factors associated with depression among heart failure patients at selected public hospitals in Addis Ababa, Ethiopia; A cross sectional study.' has been provisionally accepted for publication in PLOS Global Public Health.

Best regards,

Andreas K Demetriades, MBBChir, MPhil, FRCSEd, FEBNS.

Academic Editor

Thanks for submitting your work, and persevering with the three rounds of peer review.

Some in-house editorial assistance with the English language and grammar can hopefully help further.

I recommend for publication, as per peer review.

Reviewer Comments (if any, and for reference):

Reviewer's Responses to Questions

**Comments to the Author**

1. If the authors have adequately addressed your comments raised in a previous round of review and you feel that this manuscript is now acceptable for publication, you may indicate that here to bypass the “Comments to the Author” section, enter your conflict of interest statement in the “Confidential to Editor” section, and submit your "Accept" recommendation.

Reviewer #1: All comments have been addressed

2. Does this manuscript meet PLOS Global Public Health’s publication criteria? Is the manuscript technically sound, and do the data support the conclusions? The manuscript must describe methodologically and ethically rigorous research with conclusions that are appropriately drawn based on the data presented.

Reviewer #1: Yes

3. Has the statistical analysis been performed appropriately and rigorously?

Reviewer #1: Yes

4. Have the authors made all data underlying the findings in their manuscript fully available (please refer to the Data Availability Statement at the start of the manuscript PDF file)?

Reviewer #1: Yes

5. Is the manuscript presented in an intelligible fashion and written in standard English?

Reviewer #1: Yes

6. Review Comments to the Author

Reviewer #1: Thank you for revising the manuscript; it has improved. I have only minor comments at this stage:

- There are many grammatical and spelling errors which need to be corrected. For example, "burden" (not burdon), "severe" (not sever), in "patients" (not patient) with CVD, etc.

- References need to be standardized.

- Clarification needed on page 6 - under section 2.4 Sampling procedure and technique: "Illegibility" should be corrected to "eligibility" or the sentence be corrected to reflect this.

- Decimal places - correct to 2dp instead of 3dp for SD.

7. PLOS authors have the option to publish the peer review history of their article (what does this mean?). If published, this will include your full peer review and any attached files.

**Do you want your identity to be public for this peer review?** For information about this choice, including consent withdrawal, please see our Privacy Policy.

Reviewer #1: No
